# Identification of Vietnamese Flea Species and Their Associated Microorganisms Using Morphological, Molecular, and Protein Profiling

**DOI:** 10.3390/microorganisms11030716

**Published:** 2023-03-09

**Authors:** Ly Na Huynh, Adama Zan Diarra, Quang Luan Pham, Jean-Michel Berenger, Van Hoang Ho, Xuan Quang Nguyen, Philippe Parola

**Affiliations:** 1Aix Marseille Univ., IRD, AP-HM, SSA, VITROME, 13005 Marseille, France; 2IHU-Méditerranée Infection, 19-21 Boulevard Jean Moulin, 13005 Marseille, France; 3Institute of Malariology, Parasitology and Entomology, Quy Nhon (IMPE-QN), MoH Vietnam, Zone 8, Nhon Phu Ward, Quy Nhon 590000, Vietnam

**Keywords:** MALDI-TOF MS, molecular identification, morphological identification, fleas, flea-borne microorganisms, Vietnam

## Abstract

Fleas are obligatory blood-sucking ectoparasites of medical and veterinary importance. The identification of fleas and associated flea-borne microorganisms, therefore, plays an important role in controlling and managing these vectors. Recently, Matrix-Assisted Laser Desorption/Ionization Time-of-Flight Mass Spectrometry (MALDI-TOF MS) has been reported as an innovative and effective approach to the identification of arthropods, including fleas. This study aims to use this technology to identify ethanol-preserved fleas collected in Vietnam and to use molecular biology to search for microorganisms associated with these fleas. A total of 502 fleas were collected from wild and domestic animals in four provinces in Vietnam. Morphological identification led to the recognition of five flea species, namely *Xenopsylla cheopis*, *Xenopsylla astia*, *Pulex irritans*, *Ctenocephalides canis*, and *Ctenocephalides felis*. The cephalothoraxes of 300 individual, randomly selected fleas were tested using MALDI-TOF MS and molecular analysis for the identification and detection of microorganisms. A total of 257/300 (85.7%) of the obtained spectra from the cephalothoraxes of each species were of good enough quality to be used for our analyses. Our laboratory MALDI-TOF MS reference database was upgraded with spectra achieved from five randomly selected fleas for every species of *Ctenocephalides canis* and *Ctenocephalides felis*. The remaining spectra were then queried against the upgraded MALDI-TOF MS database, which showed 100% correspondence between morphology and MALDI-TOF MS identification for two flea species (*Ctenocephalides canis* and *Ctenocephalides felis*). The MS spectra of the remaining species (three *P. irritans*, five *X. astia*, and two *X. cheopis*) were visually generated low-intensity MS profiles with high background noise that could not be used to update our database. *Bartonella* and *Wolbachia* spp. were detected in 300 fleas from Vietnam using PCR and sequencing with primers derived from the *gltA* gene for *Bartonella* and the *16S* rRNA gene for *Wolbachia*, including 3 *Bartonella clarridgeiae* (1%), 3 *Bartonella rochalimae* (1%), 1 *Bartonella coopersplainsensis* (0.3%), and 174 *Wolbachia* spp. endosymbionts (58%).

## 1. Introduction

Fleas are considered as hosts for a wide range of human pathogens. The most severe infection by fleas is a plague that is caused by the bacterium *Yersinia pestis* [1]. Fleas are still known worldwide as important vectors of several other zoonotic pathogens, including *Rickettsia typhi*, the agent of murine typus, and *Bartonella henselae*, the agent of cat-scratch disease (CSD) [2,3].

In Vietnam, *Xenopsylla cheopis* (Rothschild, 1903) and *Xenopsylla astia* (Rothschild, 1911) fleas play an important role in the transmission of bubonic plague from rodents to other rodents and to humans [4]. The first two outbreaks of human plague occurred in Nha Trang (1898) and Saigon (1906) (now known as Ho Chi Minh City) [4]. Recently, plague foci have occasionally arisen in the Central Highlands of Vietnam, with 472 confirmed plague cases, leading to 24 deaths [5]. *Pulex irritans* (Linnaeus, 1758) is a vector of various zoonotic pathogens including plague, murine typhus, and *Rickettsia felis* infection [2]. Additionally, *P. irritans* has been described as a potential vector of *B. hensalae*, the agent of CSD and *B. quintana*, the agent of trench fever [6]. The infestation of dogs with the human flea *P. irritans* due to *Bartonella* spp. has been reported in Southeast Asia (SEA) [7]. Meanwhile, the *Ctenocephalides* species are ectoparasites with a global distribution and are vectors of various pathogens [8], many of which are also well known to infect humans [9].

Despite the reports of murine typhus, plague, rickettsial and *Bartonella* infections [5,10,11], studies on fleas and their associated microorganisms have been poorly investigated in Vietnam [12]. However, there are 51 flea species and subspecies that have been described. Among those, one new species of the *Peromyscopsylla himalaica* flea was recently found in Vietnam for the first time [13]. The accurate identification of most fleas is an essential step in studying and surveying flea-borne diseases. Undoubtedly, the list of Vietnamese flea fauna remains incomplete and is likely to be extended if further investigations on fleas are carried out [13]. However, the presence of standard taxonomic keys and reference data specific to Vietnamese flea species are currently lacking, which makes the morphological identification of fleas more difficult and sometimes nigh impossible. The identification of flea species entirely based on morphological aspects that require extensive entomological expertise, therefore, remains challenging for Vietnamese researchers. Over the last two decades, molecular approaches have been used for the identification of arthropods and their associated microorganisms. Nevertheless, these approaches have various drawbacks, including being time-consuming and costly, as well as requiring primer-specific targeting and reference sequence availability in GenBank.

The development of an alternative tool enabling the rapid, reliable, and affordable identification of fleas is, therefore, needed. Matrix-Assisted Laser Desorption/Ionization Time-Of-Flight Mass Spectrometry (MALDI-TOF MS), a useful tool that relies on the analysis of protein fingerprints, has revolutionised clinical microbiology diagnostics. More recently, MALDI-TOF MS has been used successfully to identify arthropods such as fleas [6]. Specifically, MALDI-TOF MS makes it possible to distinguish between fleas that are infected and not infected with *Borrelia* and *Bartonella* spp. [6]. This study aimed to identify flea species captured from small wild and domestic mammals in Vietnam and their associated microorganisms using morphological, molecular, and MALDI-TOF MS methods.

## 2. Materials and Methods

### 2.1. Study Sites, Flea Collection and Morphological Identification

The collection areas included four provinces: Binh Dinh (13°37′ N; 108°59′ E); Gia Lai (13°11′ N; 108°41′ E); Dak Lak (12°49′ N; 108°27′ E); and Dak Nong (12°40′ N; 107°44′ E). Field-collected flea specimens were captured between June and October 2021 in the Central and Highlands areas of Vietnam. QGIS Version 3.10 was used to build the map of Vietnam displaying the collection regions and the Vietnamese layers were downloaded from DIVA-GIS at the following link: https://www.diva-gis.org/datadown (accessed on 5 October 2022). All specimens were collected by an entomological team from IMPE-QN from the skin of rodents and domestic animals (cats and dogs) using forceps (Figure 1A,B).

The handling of wild animals in this study was carried out following the guidelines approved by the American Society of Mammalogists (http://www.jstor.org/stable/1383033, accessed on 2 March 2023) [14]. Morphologically, fleas were first identified at species level using dichotomous keys [15] by an entomological team from IMPE-QN, Vietnam. Fleas from the same host were counted and then stored in the same tube containing 70% ethanol and placed at room temperature. Flea specimens were then transferred to the Institut Hospitalo-Universitaire (IHU) Méditerranée Infection in Marseille (Marseille, France) for MALDI-TOF MS and molecular analysis. In the laboratory, all flea specimens were then morphologically verified, again by entomologists, using a magnifying stereomicroscope (Zeiss Axio Zoom.V16, Zeiss, Marly le Roi, France) and taxonomic keys [15,16]. Morphological identification was only conducted if all distinguishing features were clearly visible.

### 2.2. Flea Dissection and Specimen Preparation

Fleas were individually removed from 70% ethanol and rinsed twice in distilled water for five minutes. Each flea was dissected into three parts (cephalothorax, dorsal, and ventral part of abdomen) using a sterile surgical blade and placed in 1.5 mL Eppendorf tubes, as previously described [17]. The cephalothorax of each flea and the dorsal half of the abdomen were subjected to MALDI-TOF MS and molecular biology analyses, respectively. The ventral halves of the abdomen were frozen at −20 °C for backup.

### 2.3. DNA Extraction and Validation of Morphological Identification Using Molecular Analysis

The dorsal abdomen parts of all fleas were incubated at 56 °C overnight in 180 µL of G2 buffer (Qiagen, Hilden, Germany) and 20 µL of proteinase K (Qiagen, Hilden, Germany). DNA was individually extracted from 200 µL of the incubation solution using an EZ1 DNA tissue kit (Qiagen), according to the manufacturer’s recommendations. The eluted DNA extraction was then stored at −20 °C.

DNA samples from molecularly identified fleas were subjected to standard PCR in an automated DNA thermal cycle to amplify a 540-base pair (bp) fragment of the mitochondrial *ITS2* gene, as described previously [18]. For all specimens for which we did not obtain a sequence using the *ITS2* gene, we used amplification of a 1000 bp fragment of the mitochondrial *ITS1* gene [18]. *Ctenocephalides felis* (Bouché, 1835) DNA was utilized as a positive control, while a combination without DNA was employed as a negative control. Purified PCR products were sequenced in the same manner as previously described [18]. All sequences were clustered and processed using the ChromasPro software (Version 1.7.7) (Technelysium Pty. Ltd., Tewantin, QL, Australia), and were then compared to the reference sequences available in GenBank (http://blast.ncbi.nlm.nih.gov/ (accessed on 31 January 2023)).

### 2.4. MALDI-TOF MS Analysis

**Sample preparation** Cephalothoraxes from each flea were individually placed in 1.5-mL Eppendorf tubes and dried at 37 °C overnight. To each tube, 40 μL of high-performance liquid chromatography (HPLC) grade water was added and incubated at 37 °C overnight, as previously described [19]. The cephalothorax was ground up in a mix of 20 μL of 70% (*v*/*v*) formic acid (Sigma) and 20 μL of 50% (*v*/*v*) acetonitrile (Fluka, Buchs, Switzerland), with a small amount of 1.0 mm diameter glass beads (Sigma, Lyon, France) using a tissue layer machine (Qiagen). The cephalothorax was crushed at a frequency of 30 Hz for one minute three times, as in a previous protocol [20]. After centrifugation, 1 μL of the supernatant of each sample was spotted in quadruplicate onto a MALDI-TOF MS steel plate (Bruker Daltonics, Wissembourg, France) and overlayered after drying at room temperature with a matrix solution composed of 1 μL of saturated alpha-cyano-4-hydroxycinnamic acid (Sigma France), 50% acetonitrile (*v*/*v*), 2.5% trifluoroacetic acid (*v*/*v*) (Sigma-Aldrich Co., Ltd., Gillingham Dorset, UK), and high-performance liquid chromatography (HPLC) grade water, as previously described [21]. The target steel plate was air-dried at room temperature for a few minutes before being deposited into the Microflex LT MALDI-TOF MS apparatus (Bruker Daltonics, Germany) for analysis. The quality of the matrix, sample spotting, and operation of the MALDI-TOF MS machine were administered using the cephalothorax of a *C. felis* flea reared in our laboratory as a positive control.

### 2.5. MALDI-TOF MS Parameters

The obtained protein mass profile from the flea cephalothorax was visualised using a Microflex LT MALDI-TOF (Bruker Daltonics, Bremen, Germany) mass spectrometer with FlexControl software (Version 3.3; Bruker Daltonics), with detection in positive ion linear mode at a laser frequency of 50 Hz in a mass range of 2–20 kDa.

### 2.6. Spectral Analysis

The flexAnalysis Version 3.3 software was applied to assess spectral quality, reproducibility, and specificity. Poor quality spectra, i.e., those with low intensity (3000 AU), non-reproducibility, and background noise, were excluded from the study. By comparing the average spectral profiles (MSP, main spectrum profile) obtained from four places on each flea cephalothorax, according to species, using MALDI-Biotyper v3.0 software (Bruker Daltonics, Billerica, MA, USA), the reproducibility of MS spectra was ascertained [22]. The reproducibility and specificity of MS spectra were evaluated using gel-view, principal component analysis (PCA), and cluster analysis (MSP dendrogram). ClinProTools v2.2 with the manufacturer’s default settings was used to perform gel-view and PCA. The MSP provided by MALDI-Biotyper v3.0 software was compared to clustering based on protein mass profile (i.e., their mass signals and intensities) in the cluster analysis [22].

### 2.7. Reference Database Creation and Blind Test

The reference MS spectra were created using spectra from the extracted cephalothoraxes of each flea species using MALDI-Biotyper software v3.0. (Bruker Daltonics) [23]. MSPs were generated using an unbiased algorithm and data from peak position, intensity, and frequency [23]. MS spectra of cephalothoraxes from ten specimens of two flea species (five *C. canis* and five *C. felis*) identified morphologically and molecularly were added to our homemade MS spectra database [20]. This DB already consisted of the spectra of specimens belonging to eight flea species (*Archaeopsylla erinacei*, *C. felis*, *Ctenocephalides canis* (Curtis, 1826), *Leptopsylla taschenbergi*, *Nosopsyllus fasciatus*, *Pulex irritans*, *Stenoponia tripectinata*, and *Xenosylla cheopis*), which had been preserved in various conditions [17,20,24]. The ten MS spectra of the remaining three flea species (three *P. irritans*, five *X. astia*, and two *X. cheopis*), which were low-quality spectra, were not added to our house MS reference spectra database as reference spectra. A blind test against the updated database was performed with the remaining specimens of both *C. canis* and *C. felis* flea species. The log score values (LSVs) obtained from the MALDI-Biotyper software v.3.3, which ranged from 0 to 3, were used to estimate the reliability of species identification.

### 2.8. PCR Detection of Microorganisms in Fleas

The Eurogentec Takyon qPCR kit (Takyon, Eurogentec, Seraing, Belgium) was used in real-time PCR (quantitative PCR) according to the manufacturer’s protocol using a PCR detection system called CFX Connect™ Real-Time (Bio-Rad). The qPCR reaction contained 10 μL of Master Mix Roche (Eurogentec, Belgium), 0.5 μL of each primer probe and UDG, 3 μL of sterile distilled water, and 5 μL of the extracted DNA. DNA from flea specimens was screened to detect microorganisms using specialised primers and targeted probes, including bacteria from the Anaplasmataceae family, *Rickettsia* spp., *Borrelia* spp., *Bartonella* spp., and *Coxiella burnetii* (Table 1). DNA positive for bacteria from the Anaplasmataceae family were re-checked using *Wolbachia* spp. specific real-time PCR using the 16S rRNA gene. Negative specimens for *Wolbachia* spp. were then retested with standard PCR using the gene 23S Anaplasmataceae amplifying a 485 bp fragment. The DNA from *Rickettsia montanensis*, *Anaplasma phagocytophilum*, *Bartonella elizabethae*, *Borrelia crocidurae*, and *Coxiella burnetii* was used as a positive control, and DNA from *C. felis* breeding in our laboratory, which was free of the bacteria analysed, was used as a negative control. When the cycle threshold (Ct) was <36, the samples were considered to be positive [25].

Following qPCR, *Bartonella*-positive specimens were subjected to amplifying and sequencing of a 1000 bp fragment of the *glt*A rRNA gene. We randomly selected ten specimens that were *Wolbachia* spp.-positive following qPCR and subjected them to amplification and sequencing of a 438 bp fragment of the 16S rRNA gene.

The attained sequences of *Bartonella* spp. and *Wolbachia* spp. were clustered and analysed with the ChromasPro software (Version 1.7.7) (Technelysium Pty. Ltd., Tewantin, QL, Australia), and were then compared to the NCBI’s reference sequences database, which is available in GenBank (http://blast.ncbi.nlm.nih.gov/, accessed on 2 March 2023). The neighbour-joining (NJ) method with 1000 replicates was developed for phylogenetic tree analysis. MEGA software version 7.0 (https://www.megasoftware.net/, accessed on 2 March 2023) was used to align the DNA sequences.

## 3. Results

### 3.1. Flea Collection and Morphological Identification

A total of 502 fleas were collected from four provinces in Vietnam including 96 in Binh Dinh, 227 in Gia Lai, 35 in DakLak, and 144 in Dak Nong. Morphologically, the fleas identified belonged to 5 species (Appendix A), including 3 *X. cheopis* and 8 *X. astia* collected from rodents, 4 *P. irritans* from dogs, and 51 *C. felis* and 436 *C. canis* collected from cats and dogs. *X. cheopis* and *X. astia* fleas were captured from four species of rodents including *Rattus exulans*, *Rattus niviventer*, *Rattus norvegicus*, and *Callosciurus notatus*.

### 3.2. Molecular Identification

To confirm our morphological identification, a total of 20 flea specimens were submitted for molecular analysis using the *ITS*2 gene, including 2 specimens of *X. cheopis*, 5 *X. astia*, 3 *P. irritans*, 5 *C. felis*, and 5 *C. canis* (Table 2).

We successfully obtained nine sequences of good quality belonging to three flea species using the *ITS2* gene, namely, three specimens for *P. irritans*, five for *X. astia*, and one for *X. cheopis.* The BLAST analysis indicated that fleas morphologically identified as *P. irritans* and *X. cheopis* were 100% identical to their respective homologous sequences available in GenBank (Accession number: KX982861; DQ295095). The sequences acquired from five fleas that were morphologically classified as belonging to the *X. astia* species, however, were more closely related to the sequences that had been deposited as *X. cheopis* in GenBank, with an identity range of 83.7–84.89% (KX982860). The specimens of two flea species, *C. canis* and *Ct felis*, which could not be identified using the *ITS*2 gene, were sequenced using the *ITS*1 gene. The BLAST analysis revealed that the sequences obtained from *C. canis* matched with 100% identity with the GenBank reference sequence for *C. canis* (MT895636, HF563590) and those obtained from *C. felis* were 99.84–100% similar to their respective homologous species available in GenBank (MT895636, HF563590).

### 3.3. MS Spectra Analysis

The cephalothoraxes of 300 flea specimens, including 3 morphologically identified as *P. irritans*, 5 as *X. astia*, 2 as *X. cheopis*, 48 as *C. felis*, and 242 as *C. canis* were randomly selected for MALDI-TOF MS analysis (Table 2). The visualisation of the MS spectra obtained from the *C. canis* and *C. felis* specimens showed that 93.8% (45/48) and 87.6% (212/242) of the spectra were of high quality (peak intensity > 3000 arbitrary units (a.u.), no background noise, and baseline subtraction correct), respectively (Figure 2A and Table 2). The intraspecies reproducibility and interspecies specificity of the MS spectra of different specimens were confirmed using a dendrogram (Figure 2B), gel-view, and PCA (Figure 2C,D) analysis. According to the dendrogram, gel-view, and PCA analysis, all specimens of the same species were on the same branches or were grouped. Conversely, the remaining species’ MS spectra were visually generated low-intensity MS profiles with high background noise (Figure 2A). Hence, the spectra from *P. irritans*, *X. astia*, and *X. cheopis* were considered as of insufficient quality and were excluded from the dendrogram, gel-view, and PCA analyses.

Five representative spectra from each species were used to create the dendrogram, which showed that all of the flea specimens from the same species were grouped on the same branch to assess intraspecies reproducibility and interspecies specificity (Figure 2B).

### 3.4. MALDI-TOF MS Flea Identification

The accuracy of the MALDI-TOF MS identification of the flea specimens was assessed by querying 247 morphologically identified specimens (207 *C. canis* and 40 *C. felis*) against our upgraded MALDI-TOF MS reference database with five spectra per species, which was validated using molecular biology (Table 2).

The interrogation of the spectra of 247 flea specimens showed that all matched their counterparts in our MALDI-TOF MS database, i.e., a concordance between our morphological identification and MALDI-TOF MS. The LSVs of the *C. canis* specimens ranged from 1.708 to 2.438 (a mean = 2.024 ± 0.306), and those of *C. felis* ranged from 1.731 to 2.733 (a mean = 2.173 ± 0.303). The spectra of the flea specimens updated in the MS protein profile database were deposited on the website of the University Hospital Institute (UHI) under the following https://doi.org/10.35088/rbqp-g648, accessed on 2 March 2023.

### 3.5. Detection of Microorganisms in Fleas

The DNA of a total of 300 fleas was screened for five bacterial groups (bacteria in the Anaplasmataceae family, *Rickettsia* spp., *Borrelia* spp., *Bartonella* spp., and *Coxiella burnetii*) using qPCR. Only the DNA of the bacteria in the Anaplasmataceae family and *Bartonella* spp. were detected in 184/300 (60.3%) of our specimens. The DNA of the bacteria in the Anaplasmataceae family was detected in 174/300 (58%) of fleas (Table 3). The DNA of *Bartonella* spp. was detected in 10/300 (3%) fleas with qPCR using the *ITS2* gene (Table 3). The infected specimens included *C. canis*, *C. felis*, and *X. astia*. Notably, five *X. astia* and one *C. canis* were co-infected with both Anaplasmataceae and *Bartonella* spp. The DNA of the bacteria in *Rickettsia* spp., *Borrelia* spp., and *Coxiella burnetii* was not found in any fleas.

All 174 flea specimens that were positive for the Anaplasmataceae family were then found to be positive for *Wolbachia* spp. using the *16S* rRNA gene. We randomly selected ten positive *Wolbachia* specimens for sequencing. The analysis of the *Wolbachia 16S* rRNA fragment indicated that *Wolbachia* spp. from *X. astia*, *C. felis*, and *C. canis* were between 99.69% and 100% similar to the corresponding sequences of *Wolbachia pipientis* (MN123077, MN123078) as well as other sequences deposited in GenBank, such as *Wolbachia* endosymbionts of various arthropods (DQ399344). A total of seven out of ten *Bartonella*-positive specimens were successfully amplified with standard PCR using a targeted fragment of the *gltA* gene fragment. The BLAST analysis showed genetic distinctions in the three species of *Bartonella* spp. found in our study based on the *glt*A gene. Specifically, the BLAST analysis of the three sequences obtained from *C. canis* was between 99.2% and 100% identical to the corresponding sequences of *Bartonella clarridgeiae* (KJ170239; KY913636), and three sequences achieved from *X. astia* were 100% similar to the corresponding sequences of *Bartonella rochalimae* (FN645459). One sequence obtained from *X. astia* was 100% identical to the corresponding sequence of *Bartonella coopersplainsensis* (EU111803) (Table 4).

Two phylogenetic tree analyses of *Wolbachia* spp. and *Bartonella* spp. were generated from the *16S* rRNA and *gltA* gene sequences of our amplicons, respectively. These phylogenetic trees indicated that the detected microorganisms were close to their homologues available in GenBank (Figure 3 and Figure 4).

## 4. Discussion

The fleas analysed in our study were collected mainly from cats, dogs, and small wild rodents. These ectoparasites are well known to be vectors of human and veterinary pathogens, which are considered important for public health worldwide.

In our study, the morphological identification of Vietnamese flea species revealed the presence of five species, namely, *X. astia*, *X. cheopis*, *P. irritans*, *C. canis*, and *C. felis*. All of these species were previously discovered in Vietnam and several countries in Southeast Asia (SEA), including Laos, Thailand, and Malaysia [32,33,34,35,36,37]. *X. cheopis* was the species most predominantly captured from wild rodents in Vietnam [5]. *X. cheopis* is best known for being a vector in the transmission of *Y. pestis*, the bacterial agent of the bubonic plague primarily responsible for two pandemics that marked human history in Vietnam [4] and that still poses a threat to public health in endemic countries [38,39]. *X. astia* is also included here because it has been identified in other plague-hit countries [40,41]. In Vietnam, the presence of *X. cheopis* and *X. astia* plague vectors co-exist and parasitise in commensal rodents living inside or outside dwellings but also in open biotopes (agricultural areas, savanna grasslands) and forests [33].

*Ctenocephalides canis* was the dominant flea species infection in 86.9% of both dogs and cats, followed by 10.2% *C. felis*, as already reported worldwide, including SEA [7,42]. These species are known as competent vectors of zoonotic pathogens such as *Rickettsia felis* and *Bartonella* spp. [43]. *C. felis* is the most well-known vector of *Rickettsia felis*, a causative agent of the spotted fever group rickettsiosis [44]. In SEA, the first human case of a *R. felis* infection was identified on the Thai–Myanmar border, as described in Parola et al. [45]. Since then, many human infections have been reported in Laos [46], Indonesia [47], and Vietnam [10]. Several studies have shown *R. felis* in *C. felis* fleas in Vietnam [7,12]. However, no *Rickettsia*-positive fleas were found in our specimens. Similarly, no *C. felis* analysed showed evidence of the carriage of *Bartonella* bacteria in our fleas. In contrast, we reported that three *C. canis* dog fleas were found to be infected with the *B. clarridgeiae* species, which is the causative agent of CSD in humans. This bacterium was the most common species found in cat fleas and was also detected in humans, cats, and dogs [48,49]. The human flea, *P. irritans*, was collected from 0.8% (4/502) of the dogs in our study. *P. irritans* is widespread globally and has also been detected in other wild animals (birds, rodents, bats, carnivores, and ungulates) [50,51]. Furthermore, this flea plays a role in the transmission of *Y. pestis* between humans [50,52]. However, it has also been identified as a vector of several bacterial pathogens, such as *Rickettsia* and *Bartonella* species [53]. Nevertheless, no evidence of *Bartonella* was found in the *P. irritans* fleas in our study.

Molecular biology was used to confirm the morphological identification of five flea species collected in Vietnam, namely, *X. cheopis*, *X. astia*, *P. irritans*, *C. canis*, and *C. felis*, using either the *ITS1* or *ITS2* gene, homologous sequences of which were available in GenBank. The *ITS1* gene was chosen to differentiate *C. felis* from *C. canis* because this specific marker shows a unique divergence compared with other genes frequently used to identify arthropods [54]. Nevertheless, the reference DNA barcode sequences deposited in GenBank miss the *ITS1* region or have only recently been updated [54,55], which shows the limited range of molecular technology in the choice of targeted sequences. Furthermore, one species, morphologically identified as *X. astia*, did not match the molecular identification result. This discrepancy was due to the lack of DNA sequence information for *X. astia* species in the GenBank database, which is also one of the described drawbacks of molecular biology [56].

MALDI-TOF MS has revolutionised clinical microbiology diagnostics as a result of its advantages in the routine identification of bacteria, archaea, and fungi [57]. Recently, MALDI-TOF MS has emerged as an efficient approach to the rapid and accurate identification of arthropod vectors, including fleas [6,20,55]. In our study, the high-quality spectra of *C. canis* and *C. felis* were 87.6% and 93.8%, respectively. Unfortunately, it was not possible to obtain good quality spectra from three flea species (corresponding to the three *P. irritans* samples from dogs, five *X. astia*, and two *X. cheopis* from rodents) in order to update the database and then identify using a blind test. The theories were excluded from our results, for example: (1) the specimens were stored in ethanol for a long time, while all of our samples were preserved and subsequently analysed for a period of between two and six months. Ethanol is widely used for the conservation and transportation of arthropods under field conditions because it is less restrictive than freezing [58]. Several studies have shown the efficacy of MALDI-TOF MS for the identification of arthropods preserved in ethanol for between two and ten years [58,59] and even up to several decades [19]; (2) Previous data indicated that the choice of the compartment chosen for the MALDI-TOF MS analysis, the cephalothorax, had high-quality spectra compared to different parts of the body [6,17,20]. The limited sample size for each species (between two and five specimens per species) might explain our spectra results.

We report, for the first time, the presence of *Wolbachia* spp. in three flea species from Vietnam. *Wolbachia* is a genus of bacterial endosymbiont that is known to infect both nematodes and arthropods [60]. These bacteria have been reported to enable transmission from parents to their descendants and may alter the biology, ecology, and evolution of its hosts by acting on feminisation, parthenogenesis, male-killing in the arthropods, and hence cause the cytoplasmic incompatibility of spermatozoa [60]. In Vietnam, the molecular detection of *Wolbachia* in natural mosquito populations and other arthropods has not been recorded. However, information on the detection of *Wolbachia* spp. in arthropods, including fleas found in the tropical regions of Laos, Thailand, and Malaysia, has been published [61]. In our study, 58% of the fleas were positive for *Wolbachia* spp. endosymbiont DNA. The prevalence of *Wolbachia* found in our work is higher than that identified in a study of cat fleas in France [29], although similar results on the occurrence of the *Wolbachia* infection has been established in Malaysia [61].

*Bartonella* species are intracellular parasites of erythrocytes in a wide range of various mammalian and ectoparasite hosts [62]. Most of these bacterial species have been detected in their arthropod vectors, including fleas, ticks, and lice [6,63,64], although the majority of these arthropods’ roles as vectors have yet to be proven [65]. Herein, we showed the presence of two human pathogens of the *Bartonella* genus (*B. clarridgeiae* and *B. rochalimae*) [66] and one *B. coopersplainsensis* in an endemic Australian rat [67]. The occurrence of *Bartonella* from rodent and cat fleas has been reported in China [66], Thailand [68], Japan [69], Nepal [70], Indonesia [71], and Cambodia [72]. Our results provide the first molecular evidence for *Bartonella clarridgeiae*, *B. coopersplainsensis*, and *B. rochalimae* in rodent and dog fleas from Vietnam and suggest their related flea-borne diseases.

*Bartonella clarridgeiae* has been suspected to be an additional agent of CSD, and its pathogenic role in humans has been demonstrated in Ireland [73] and the United States [74]. *B. clarridgeiae* is also known as a veterinary pathogen associated with disease in cats and dogs [75]. *Bartonella* spp. (*B. elizabethae*, *B. rattimassiliensis*, *B. tribocorum*, *B. coopersplainsensis*, and *B. queenslandensis*) have already been reported from rat mites in the Mekong Delta [76] and blood samples from rodents and bats in Vietnam [11,77]. However, the *Bartonella* genus, especially *B. clarridgeiae*, was first discovered in *C. canis* dog fleas in our study. It has been previously found in fleas in Southeast Asia (SEA) from the Thai–Myanmar border [78], Laos [32], the Philippines [79], and Indonesia [80].

We found that 1% of *X. astia* fleas, which parasitise small wild rodent species, harbour *B. rochalimae* DNA using a sequence analysis based on the *gltA* gene. *B. rochalimae* is the causative agent of CSD, which can be transmitted to humans from fleas and mites [81]. Several cases of human infection have been described in the United States [82]. A *B. rochalimae* infection was first reported in a dog from California [83]. Wild carnivores such as coyotes, foxes, and skunks, have been suggested to be major reservoirs for *B. rochalimae* in nature [81]. In Asia, *B. rochalimae* has been found in wild rodents near the China–Kazakhstan border [66].

We showed, for the first time, the prevalence of *B. coopersplainsensis* in *X. astia* rodent fleas captured in Vietnam. This *Bartonella* bacterium has not yet been described in humans [62]. However, *B. coopersplainsensis* has been detected in rats collected from SEA [72,84], Brazil [62], and New Zealand [85], and from other wild rodents in China [86] and Lithuania [87]. In Vietnam, *B. coopersplainsensis* has already been detected in trombiculid mites in rats and their reservoir host’s blood samples [76].

The *Bartonella* species detected in our specimens were *B. clarridgeiae*, *B. rochalimae*, and *B. coopersplainsensis*, with no evidence of other species such as *Bartonella* spp. (*B. rattimassiliensis*, *B. tribocorum*), being reported in the SEA region. Our results demonstrate that *X. astia* and *C. canis* fleas are potential vectors of *B. clarridgeiae* and *B. rochalimae*, which might play a role in human infection in the Central Highlands of Vietnam. 

## 5. Conclusions

Our study shows that the MALDI-TOF MS tool can be used for the rapid identification of Vietnamese flea species preserved in 70% ethanol for more than a year. However, further studies are needed to prove its effectiveness on a large number of different species. It has also provided evidence of dog and rodent fleas being the vectors for carrying *Bartonella* species in Vietnam. The detection of these pathogenic bacteria in their ectoparasites may be applied to epidemiologic surveillance and prevention strategies. Hence, further studies will be needed to identify the potential factors of *B. clarridgeiae*, *B. rochalimae*, and *B. coopersplainsensis* and to investigate whether small wild rodents and domestic dogs, as well as ectoparasite vectors, are of zoonotic importance.

## Figures and Tables

**Figure 1 microorganisms-11-00716-f001:**
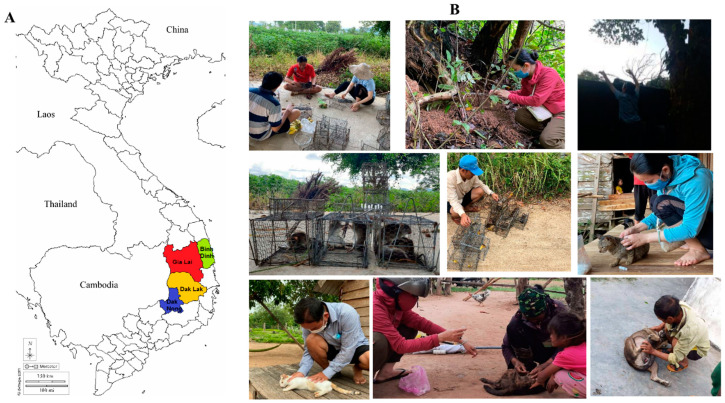
(**A**) Map of Vietnam showing the flea collection sites. (**B**) Field photographs of the wild rodents and domestic animal sampling collection beginning with type of trap used, trap preparation, trap setting, and pet owners’ support.

**Figure 2 microorganisms-11-00716-f002:**
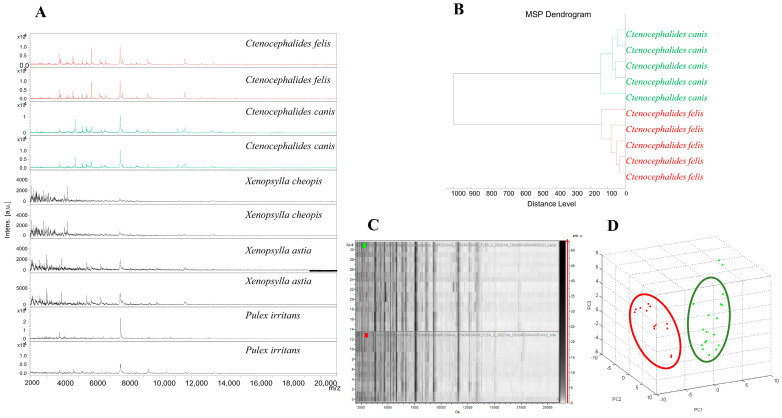
Building the database using comparison of MS spectra obtained from diverse flea species. Spectral alignment of five flea species using flexAnalysis software to show discriminative peaks; representative MALDI-TOF MS spectra from the cephalothorax of *X. cheopis*, *X. astia*, and *P. irritans* with low-intensity MS profiles and high background noise; representative MS spectra of *C. canis* and *C. felis* with high-quality spectra (peak intensity >3000 arbitrary units (a.u.), no background noise with baseline subtracted (**A**). Dendrogram of MALDI-TOF MS spectra of flea species collected in Vietnam. Biotyper software v.3.0 was used for cluster analysis (**B**). As confirmed with PCA, the MS spectra of various specimens demonstrated intraspecies reproducibility and interspecies specificity (**C**,**D**).

**Figure 3 microorganisms-11-00716-f003:**
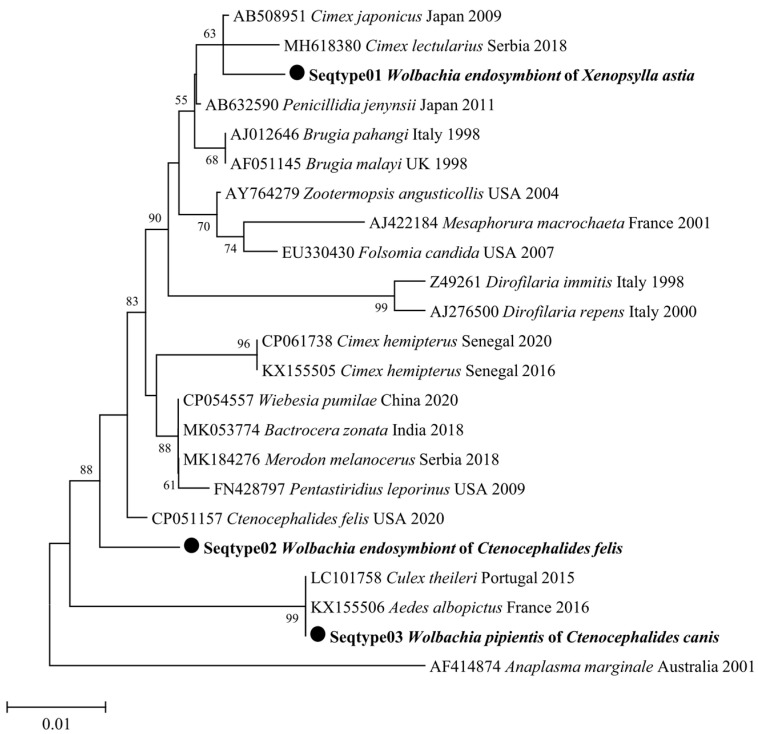
Neighbour-joining (NJ; 500 bootstrap replicates) phylogenetic tree of the *16S rRNA* gene. *Wolbachia* spp. (●) collected from *Xenopsylla astia*, *C. felis*, and *C. canis*. *Wolbachia* strains are designated following the names of their host species.

**Figure 4 microorganisms-11-00716-f004:**
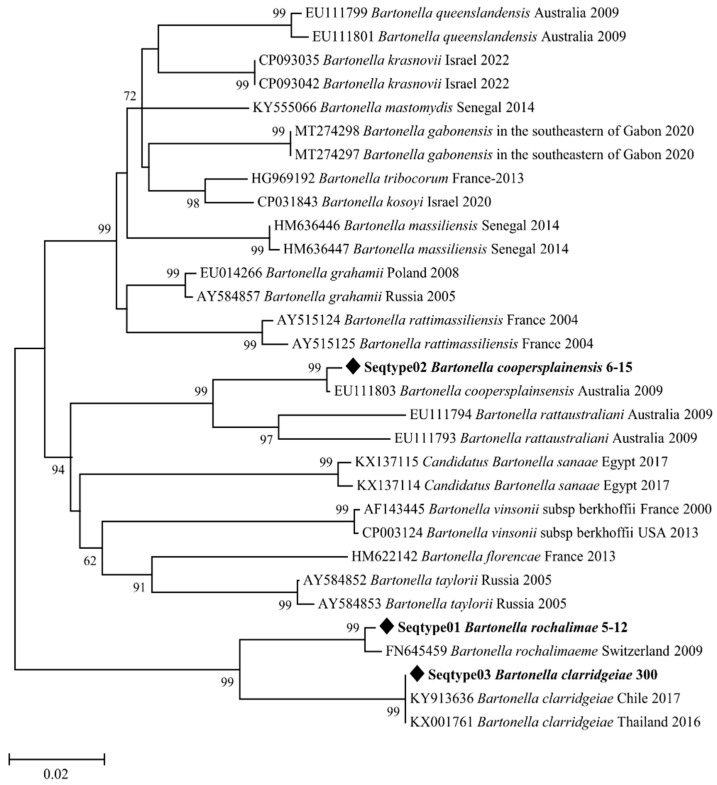
Neighbour-joining (NJ; 500 bootstrap replicates) phylogenetic tree of the *gltA* gene. *Bartonella* spp. sequences (◆) obtained in this study.

**Table 1 microorganisms-11-00716-t001:** Sequences of the primer sets used for fleas and flea-borne pathogen detection using qPCR and standard PCR.

Microorganisms	Targeted Sequence/Amplicon Size (bp)	Primers (5′-3′) and Probes (Used for qPCR Screening or Sequencing)	References
Anaplasmataceae	*23S*	f_TGACAGCGTACCTTTTGCAT r_GTAACAGGTTCGGTCCTCCA p_6FAM-GGATTAGACCCGAAACCAAG	[26]
	*23S* (485)	f_ATAAGCTGCGGGGAATTGTC r_TGCAAAAGGTACGCTGTCAC	
*Rickettsia* spp.	*gltA*(RKND03)	f_GTGAATGAAAGATTACACTATTTAT r_GTATCTTAGCAATCATTCTAATAGC p_6FAM-CTATTATGCTTGCGGCTGTCGGTTC	[27]
*Borrelia* spp.	*ITS4*	f_GGCTTCGGGTCTACCACATCTA r_CCGGGAGGGGAGTGAAATAG p_TGCAAAAGGCACGCCATCACC	[28]
*Bartonella* spp.	*ITS2*	f_GATGCCGGGGAAGGTTTTC r_GCCTGGGAGGACTTGAACCT p_GCGCGCGCTTGATAAGCGTG	[29]
	*gltA* (1000 bp)	f_ACGTCGAAAAGAYAAAAATG r_GTAATRCCAGAAATARAAATC	
*Coxiellia burnetii*	*IS30A*	f_CGCTGACCTACAGAAATATGTCC r_GGGGTAAGTAAATAATACCTTCTGG p_CATGAAGCGATTTATCAATACGTGTATG	[30]
*Wolbachia* spp.	*23S* rRNA	Wol-301-f (5′-TGGAACTGAGATACGGTCCAG-3′) Wol-478-r (5′-GCACGGAGTTAGCCAGGACT-3′) Wol-347-p (6FAM-AATATTGGACAATGGGCGAA)	[31]
	*16S* rRNA (rrs)	W-Spec_f (5′-CATACCTATTCGAAGGGATAG-3′) W-Spec_r (5′-AGCTTCGAGTGAAACCAATTC-3′)	[26]

**Table 2 microorganisms-11-00716-t002:** The number of flea species analysed for protein profiling, development of the MS reference spectra, and validation of molecular biology.

Morphological Identification	N° Tested MALDI-TOF MS/Collected	N° Obtained/Tested and MolecularID* (%identity; GenBank Accession Number)	N° of Good Spectra/ Tested	N° of Spectra Added to DB^&^	MALDI-TOF MS ID^*^ (Number Identified)	LSVs^$^ (Low–High)
*Pulex irritans*	3/4	3/3 *Pulex irritans* (99–100%; KX982861)	0/3	0	Not applicable	Not applicable
*Xenopsylla cheopis*	2/3	1/2 *Xenopsylla cheopis* (100%; DQ295059)	0/2	0	Not applicable	Not applicable
*Xenopsylla astia*	5/8	5/5 X*enopsylla cheopis* (83.16–90%; KX982860	0/5	0	Not applicable	Not applicable
*Ctenocephalides felis*	48/51	5/5 *Ctenocephalides felis* (100%; MT895636, HF583247)	45/48	5	*Ctenocephalides felis* (40)	1.731–2.733
*Ctenocephalides canis*	242/436	5/5 *Ctenocephalides canis* (100%; MH895642)	212/242	5	*Ctenocephalides canis* (207)	1.708–2.438
5 flea species	300/502		257/300			

ID*; identification, DB^&^; database; LSVs^$^; log score values.

**Table 3 microorganisms-11-00716-t003:** Microorganisms detected using real-time PCR in fleas.

Microorganisms Tested	Flea Species			
	*X. astia*	*C. canis*	*C. felis*	Total (%)
Anaplasmataceae	5	148	21	174 (58%)
*Bartonella* spp.	4	10	-	7 (2.3%)
*Rickettsia* spp.	-	-	-	-
*Borrelia* spp.	-	-	-	-
*Coxiella burnetii*	-	-	-	-

**Table 4 microorganisms-11-00716-t004:** Microorganisms detected using sequencing in fleas.

Microorganisms Tested	Flea Species				
	Per Ident (%)	*X. astia*	*C. canis*	*C. felis*	Total (%)
*Wolbachia endosymbiont*	99.69–100	3		2	NA
*Wolbachia pipiens*			5	-	NA
*Bartonella clarridgeiae*	99.2–100	-	3	-	3 (1%)
*Bartonella rochalimae*	100	3	-	-	3 (1%)
*Bartonella coopersplainsensis*	100	1	-	-	1 (0.3%)

NA; Not applicable.

## Data Availability

Not applicable.

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
