# Peer review of "Identification of Vietnamese Flea Species and Their Associated Microorganisms Using Morphological, Molecular, and Protein Profiling"

_microorganisms, 2023, doi:10.3390/microorganisms11030716_

Round 1
Reviewer 1 Report
The data obtained are interesting for a fast and reliable determination of ectoparasites, and their microorganisms in this case of fleas, by means of three different methods, as well as the generation of a reference database for one of the methods.
The authors should review the suggestions and corrections made in the document.
Put the name and year of the author of the species when it is mentioned for the first time
Some of the photos of the fleas could be improved, they look blurry
Author Response
Authors’ response-Manuscript ID: microorganisms-2220330
Title: Identification of Vietnamese flea species and their associated microorganisms using morphological, molecular and protein profiling
Corresponding author: Philippe PAROLA
Journal: Microorganisms
Reviewer 1
Comments and Suggestions for Authors
The data obtained are interesting for a fast and reliable determination of ectoparasites, and their microorganisms in this case of fleas, by means of three different methods, as well as the generation of a reference database for one of the methods.
The authors should review the suggestions and corrections made in the document.
Put the name and year of the author of the species when it is mentioned for the first time
Authors’ response: Dear reviewer, as requested, we added the name and year of the author of the species when it was mentioned for the first time in lines 47-48, 52, 138, and 188-189.
Some of the photos of the fleas could be improved, they look blurry
Authors’ response: These fleas were collected in Vietnam and stored in 70% ethanol for over 1 year. They were then rinsed twice in distilled water for five minutes before mounting. To could be easily identified by the shape of the spermatheca, we had to do mounting and took photos of them under a magnifying glass. Especially, the specimens were put in the photos that had been confirmed by molecular. The remaining specimens were used for analysing MALDI-TOF MS and molecular. So, we are unable to improve them as well as make new ones.
Additionally, these photos of the fleas were referred to Supplementary Figs 1 and 2 as requested by reviewer 2.

Reviewer 2 Report
This research article provides the morphological, molecular and MALDI-TOF MS identification of flea species. Also the identification of microorganisms associated with fleas. They identified two out of five flea species with MALDI-TOF MS. They obtained sequences of specific flea species and uploaded its to GenBank. They detected Bartonella clarridgeiae, B. rochalimae, B. coopersplainensis and Wolbachia endosymbiont in tested fleas. The importance of this study relies on the fact that such type of work is being carried out for the first time in Vietnam. The manuscript is well-structured, and it is easy to read. However, there are some issues that should be improved.
Improvement suggestions:
Line 58: delete "and": "... plague, and rickettsial and bartonella infections...".
Line 80: Borrelia in fleas? Make sure it's true. As far as I know Borrelia is more associated with ticks.
Lines 86-89: don't repeat word "province" after each location. Rewrite this sentence that the collection date would be as separate sentence.
Line 92: What species of rodents were studied for fleas?
Lines 93-108: Figure 1 B is not informative. I would recommend to remove it.
Line 117: Was a magnifying glass really used for morphological identification of fleas? May be it was done by microscope?
Lines 215-217: Remove a sentence "The primers and probes used for real-time quantitative and standard PCR in this study are listed in Table 1." because the same idea is written in 201-203 lines.
Table 1: check background color. It seems that sequences of primers of Bartonella spp. gltA are highlighted.
Figure 2 and Figure 3: it is not your result but only the repetition of identification keys. I suggest moving these figures (2 and 3) to the supplementary file.
Lines 302-339: use the same format for referring to the figure in the text.
Lines 465-466: rewrite sentence "Furthermore, this flea plays a role in domestic human-to-human transmission of Y. pestis...". For example, " Furthermore, this flea plays a role in the transmission of Y. pestis between people...."
Lines 542-545: check the number of reference for countries.
Lines 552-553: the sentence "Our work proves that MALDI-TOF MS as an efficient tool for the rapid and accurate identification of the ethanol-stored flea species." is too strong. Although you analyzed a considerable number of fleas, only two species (Ct.canis and Ct. felis) from five showed good results. Definitely, more studies on this issue are needed. So, please, avoid such strong affirmation.
References: you should select the most useful ones and eliminate the irrelevant ones
Author Response
Authors’ response-Manuscript ID: microorganisms-2220330
Title: Identification of Vietnamese flea species and their associated microorganisms using morphological, molecular and protein profiling
Corresponding author: Philippe PAROLA
Journal: Microorganisms
Reviewer 2
Comments and Suggestions for Authors
This research article provides the morphological, molecular and MALDI-TOF MS identification of flea species. Also the identification of microorganisms associated with fleas. They identified two out of five flea species with MALDI-TOF MS. They obtained sequences of specific flea species and uploaded its to GenBank. They detected Bartonella clarridgeiae, B. rochalimae, B. coopersplainensis and Wolbachia endosymbiont in tested fleas. The importance of this study relies on the fact that such type of work is being carried out for the first time in Vietnam. The manuscript is well-structured, and it is easy to read. However, there are some issues that should be improved.
Improvement suggestions:
Line 58: delete "and": "... plague, and rickettsial and bartonella infections...".
Authors’ response: Dear reviewer, as requested, we removed a letter“and” that was put above “rickettsial” in line 59
Line 80: Borrelia in fleas? Make sure it's true. As far as I know Borrelia is more associated with ticks.
Authors’ response: The reviewer is right, we have corrected this sentence in lines 80-81 as “MALDI-TOF MS makes it possible to distinguish between fleas which are infected and not infected by Bartonella spp.”
Lines 86-89: don't repeat word "province" after each location.
Authors’ response: As requested by the reviewer, we removed “province” after each location in lines 86-87.
Rewrite this sentence that the collection date would be as separate sentence.
We also rewrote this sentence as follows: “Field-collected fleas specimens were captured between June and October 2021 in the Central and Highlands areas of Vietnam” in lines 87-89
Line 92: What species of rodents were studied for fleas?
Authors’ response: We thank the reviewer, the rodent species studied for the fleas have been added to the manuscript as follows in lines 233-234: “Fleas (Xenopsylla cheopis and Xenopsylla astia) were captured from 4 species of rodents as Rattus exulans, Rattus niviventer, Rattus norvegicus, and Callosciurus notatus. ”
Lines 93-108: Figure 1 B is not informative. I would recommend to remove it.
Authors’ response: In figure 1B, we would show the procedure of field sampling collection for rodents and domestic animals beginning with trap-used type, trap preparation, and trap setting. The type of traps was used as live animal traps. For dogs and cats, we obtained collaboration from the owners. So, we would like to keep figure 1B and modify the legend of figure 1B in lines 107-109 as follows: “Field photographs of the wild rodents and domestic animal sampling collection beginning with trap-used type, trap preparation, trap setting, and pet owners’ support”.
Line 117: Was a magnifying glass really used for morphological identification of fleas? May be it was done by microscope?
Authors’ response: We thank the reviewer, Yes a magnifying-stereomicroscope connected to a computer for a better visualization of morphological characteristics was used and not a microscope. We have changed a magnifying glass by a magnifying-stereomicroscope in line 118.
Lines 215-217: Remove a sentence "The primers and probes used for real-time quantitative and standard PCR in this study are listed in Table 1." because the same idea is written in 201-203 lines.
Authors’ response: As requested by the reviewer, we have removed this sentence
Table 1: check background color. It seems that sequences of primers of Bartonella spp. gltA are highlighted.
Authors’ response: The reviewer is right, we cleaned them in Table 1
Figure 2 and Figure 3: it is not your result but only the repetition of identification keys. I suggest moving these figures (2 and 3) to the supplementary file.
Authors’ response: As requested by the reviewer, we moved the figures (2 and 3) to the supplementary Fig1 and Fig2, respectively.
Then, Figures 4, 5, and 6 were named Figures 2, 3, and 4, respectively
Lines 302-339: use the same format for referring to the figure in the text.
Authors’ response: we corrected the format of each figure throughout the manuscript
Lines 465-466: rewrite sentence "Furthermore, this flea plays a role in domestic human-to-human transmission of Y. pestis...". For example, " Furthermore, this flea plays a role in the transmission of Y. pestis between people...."
Authors’ response: As requested by the reviewer, we changed this paragraph as follows: “Furthermore, this flea plays a role in the transmission of Y. pestis between humans to humans” in lines: 425-426.
Lines 542-545: check the number of reference for countries.
Authors’ response: We checked the number of references for countries. All are fine
Lines 552-553: the sentence "Our work proves that MALDI-TOF MS as an efficient tool for the rapid and accurate identification of the ethanol-stored flea species." is too strong. Although you analyzed a considerable number of fleas, only two species (Ct.canis and Ct. felis) from five showed good results. Definitely, more studies on this issue are needed. So, please, avoid such strong affirmation.
Authors’ response: As requested by the reviewer, we modified this sentence in line 512-514: “Our work shows that the MALDI-TOF MS tool can be used for rapid identification of Vietnamese flea species stored in 70% ethanol for more than a year. However, further studies are needed to prove its effectiveness on a large number of different species.”
References: you should select the most useful ones and eliminate the irrelevant ones
Authors’ response: As requested by the reviewer, several references were filtered and arranged in order.
